# Short-term availability of adult-born neurons for memory encoding

Jérémy Forest[1,2,3], Mélissa Moreno[1,2,3], Matthias Cavelius[1,2], Laura Chalençon[1,2], Anne Ziessel[1,2], Joëlle Sacquet[1,2], Marion Richard[1,2], Anne Didier[1,2,4] & Nathalie Mandairon [1,2,4]*

Adult olfactory neurogenesis provides waves of new neurons involved in memory encoding. However, how the olfactory bulb deals with neuronal renewal to ensure the persistence of pertinent memories and the flexibility to integrate new events remains unanswered. To address this issue, mice performed two successive olfactory discrimination learning tasks with varying times between tasks. We show that with a short time between tasks, adult-born neurons supporting the first learning task appear to be highly sensitive to interference. Furthermore, targeting these neurons using selective light-induced inhibition altered memory of this first task without affecting that of the second, suggesting that neurons in their critical period of integration may only support one memory trace. A longer period between the two tasks allowed for an increased resilience to interference. Hence, newly formed adult-born neurons regulate the transience or persistence of a memory as a function of information relevance and retrograde interference.

---

[1] INSERM, U1028; CNRS, UMR5292; Lyon Neuroscience Research Center, Neuroplasticity and Neuropathology of Olfactory Perception Team, Lyon F-69000, France. [2] Claude Bernard University-Lyon1 and University of Lyon, Lyon F-69000, France. [3]These authors contributed equally: Jérémy Forest, Mélissa Moreno. [4]These authors jointly supervised this work: Anne Didier, Nathalie Mandairon. *email: nathalie.mandairon@cnrs.fr

All animals are exposed to a changing olfactory environment requiring constant adjustments of odor perception and memory, in order for the olfactory system to efficiently guide the animal's behavior. Adult-born neurons provided by neuroblasts formed in the subventricular zone[1–4], constantly integrate the olfactory bulb (OB) and underlie olfactory learning[5–14]. However it has been proposed that, in the hippocampus, adult neurogenesis promotes forgetting. Indeed, an increase in hippocampal neurogenesis tends to enhance memory clearance thereby reducing proactive interference while decreased neurogenesis prevents forgetting and reduces behavioral flexibility[15–17]. Thus, the neuronal turnover provided by adult neurogenesis could sub-serve memory formation and regulate the transience of the memory trace. This raises the issue of how this 'remembering/forgetting' balance is regulated as a function of environmental demand, allowing the animal to keep relevant information in memory for optimal behavioral adaptation. However, the parameters determining memory persistence or forgetting such as information relevance or the time between successive inputs and to what extent they depend on adult-born neurons are unknown. More specifically, adult-born neurons show a critical period during which they can be recruited by learning[6] but whether they can support successive memory traces within this critical period and whether they suffer from memory interference are unknown.

Here, using the olfactory system in mice, we tackle the issue of how the acquisition of new memories influences the information already stored in the network and we identify the role of adult-born neurons in these processes. We used perceptual olfactory discrimination learning, which depends on newly formed neurons in the OB[8] to show that (1) adult-born neurons saved by learning are present in the OB as long as the task is remembered; (2) with a second learning task occurring soon after the first, introducing retrograde interference, the new memory overwrites the older one and alters survival of previously recruited adult-born neurons unless (3) the first learned odorants are maintained in the environment. Finally, using sequential labeling of adult-born neurons and selective optogenetic inactivation, we showed (4) that each successive learning is supported by a specific population of adult-born neurons.

## Results

**Performances are linked to adult-born neurons fate.** Previous studies have reported increased survival of adult-born granule cells after perceptual discrimination learning[8]. To better understand the potential role of these neurons in long-term memory, we first asked whether increased bulbar neurogenesis persists for as long as the learned information is remembered. Thus, different groups of mice underwent perceptual discrimination learning consisting in a 10-day enrichment period with (+)limonene and (−)limonene (+lim and −lim), odorants which are not spontaneously discriminated by mice. Their performance in discriminating these two odorants was then analyzed from 1 to 5 weeks post learning (T1, T2, T3, T4, and T5, Fig. 1a) using a habituation/dishabituation test. In this test, mice were exposed to the first odorant of the pair 4 times (Hab1 to Hab4; inter-trial interval 5 min) followed by exposure to the second odorant of the pair (Test, 5 min between Hab4 and Test). The time spent by the mouse investigating the odorant was recorded for all trials. Decreased investigation time across habituation trials reflects habituation to the odorant used and a longer time spent investing Test than Hab4 shows discrimination of the two odorants of the pair. At all time-points analyzed, mice exhibited habituation behavior (Supplementary Table 1, Supplementary Fig. 1). Regarding discrimination, using a two-way ANOVA, we observed significant differences between groups (NE: Non Enriched, T1, T2, T3, T4, T5: $F_{(5,106)} = 2.39$; $p = 0.043$) and effect of the trial (Hab4 vs Test,

$F_{(1,106)} = 9.89$; $p = 0.002$). Enriched animals were able to discriminate the two enantiomers of limonene from T1 to T3 (Paired $t$-test, $p < 0.05$ between Hab4 and Test) but not after the longer time periods T4 and T5 (Paired $t$-test, $p > 0.05$ between Hab4 and Test; Supplementary Table 1, Fig. 1b). To assess the specificity of learning, discrimination of another pair of similar odorants, decanal/dodecanone (dec/dodec), to which the animals had not been enriched was also tested. No discrimination of dec and dodec was seen after enrichment with +lim and −lim, (ANOVA group effect: $F_{(5,90)} = 1.18$, $p = 0.32$; trial effect: $F_{(1,90)} = 0.05$; $p = 0.83$, $p > 0.05$ between Hab4 and Test) (Supplementary Table 1; Fig. 1c). NE control animals did not discriminate any of the two odorant pairs (Paired $t$-test, T1: $p > 0.05$ between Hab4 and Test; Supplementary Table 1; Fig. 1b, c).

To assess adult-born neuron survival, a cohort of adult-born neurons present in the OB at the beginning of learning was labeled by injecting the DNA synthesis marker 5′-bromo-2′-deoxyuridine (BrdU) 8 days before learning[8,9]. BrdU-positive cells were counted in the granule cell layer for each time period post-learning (T1 to T5). We did not assess neurogenesis in the glomerular cell layer of the OB since we had previously found no modulation of the rate of neurogenesis in this layer after perceptual learning[8]. Analysis was performed based on learning performance, by comparing animals who discriminated after learning (T1, T2, and T3) to those who did not (T4 and T5) and to their corresponding NE groups (ANOVA, $F_{(3, 32)} = 4.1$, $p = 0.014$; Fig. 1d and Supplementary Fig. 2). We observed a higher density of BrdU-positive cells in enriched groups able to discriminate the two odorants used for enrichment compared to enriched groups unable to do so (T1-T2-T3 versus T4-T5, $t$-test, $p = 0.015$, Fig. 1d) and compared to NE groups (enriched T1-T2-T3 vs non enriched T1-T2-T3, $t$-test, $p = 0.028$, Fig. 1d and Supplementary Fig. 2). Accordingly, the density of BrdU-positive cells was highly correlated to the index of discrimination (Pearson correlation: $R = 0.84$; $p = 0.038$) (Fig. 1e). The density of BrdU-positive cells showed no decline over the time points in NE animals (ANOVA, $F_{(4,11)} = 0.8$, $p = 0.5$, Supplementary Fig. 2).

To determine whether the involvement of adult-born cell population in processing of the learned odorants relates to the discrimination ability, we assessed the density of cells co-expressing BrdU and Zif268 as an index of adult-born cell activation at the different time points post learning in three experimental groups. More precisely, we assessed the responsiveness of adult-born cells to +lim (odorant used for the enrichment) in enriched and NE animals. In addition, we assessed the response to dec (odorant not used for enrichment) of adult-born cells in +lim/−lim enriched animals (Fig. 1f). A 2-way ANOVA showed a group effect ($F_{(2,40)} = 33.68$, $p < 0.0001$), a time effect ($F_{(2,40)} = 14.08$, $p < 0.0001$) and an interaction ($F_{(8,40)} = 8.13$, $p < 0.0001$) indicating that the differences in BrdU/Zif268 positive cell density observed between groups depend on the time post learning. Interestingly, the density of Zif268 expression in adult-born neurons was significantly higher in enriched animals in response to +limonene at T1 to T3 compared to the other groups (Tukey tests, T1 versus T4 and T5, $p = 0.0003$; T2 versus T4 and T5, $p = 0.006$; T3 versus T4, $p = 0.0005$ and T3 versus T5, $p = 0.0006$). This result is in accordance with the performance of discrimination (Fig. 1f). Regarding the NE group, no time effect was observed (ANOVA, $F_{(4,15)} = 4.66$, $p = 0.84$). These changes were underlined not only by changes in the density of BrdU positive cells but also by changes in percentage of BrdU/Zif268 positive cells (Supplementary Fig. 3). Notably, a higher percentage of BrdU/Zif268 was found in enriched animals at T1-T3 compared to T4-T5 ($t$-test, $p = 0.00022$) in response to +lim but not dec, indicating of a higher involvement of BrdU positive cells in processing the learned

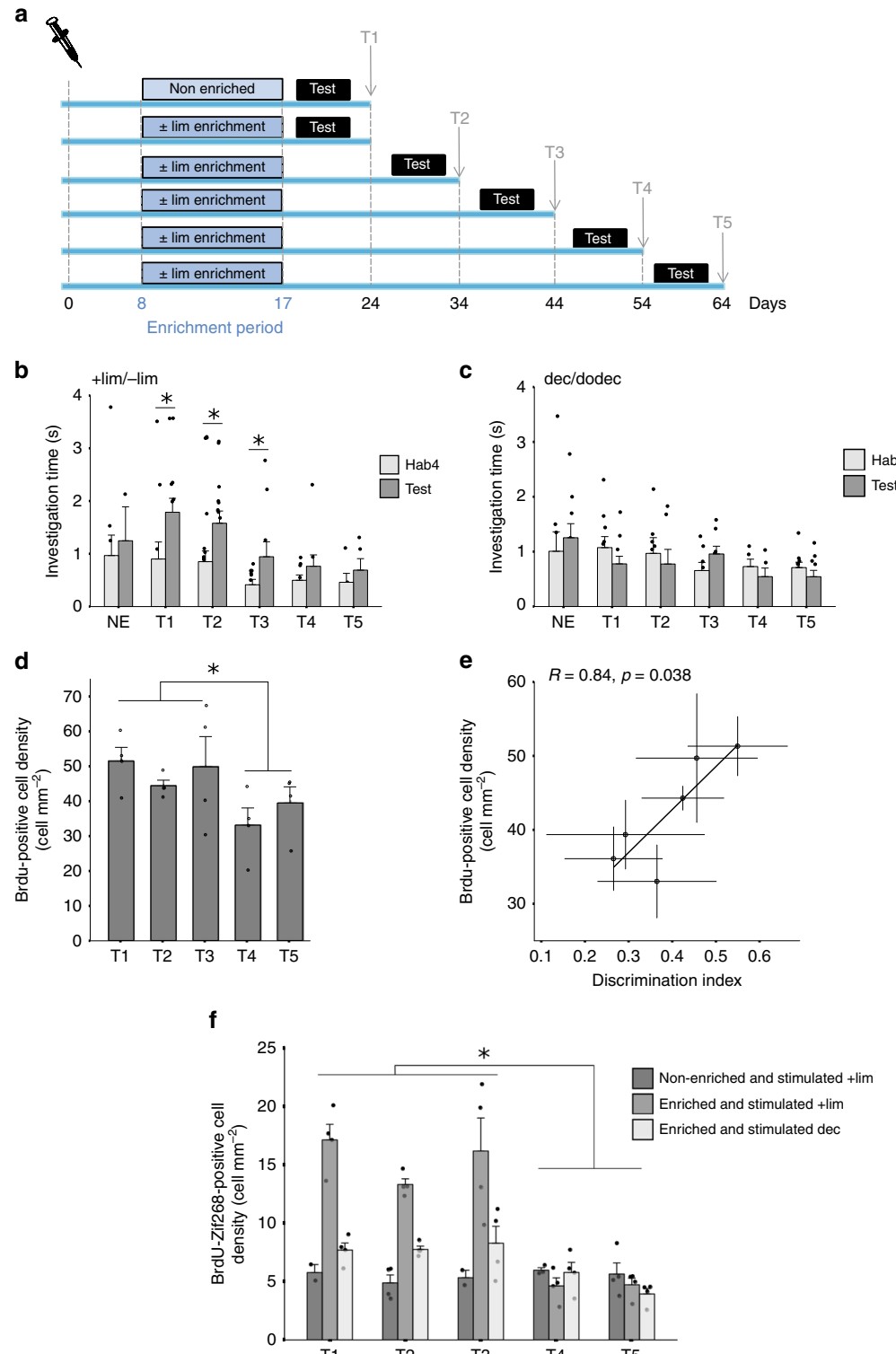

**Fig. 1 The presence of adult-born neurons correlates with mnesic performance. a** Experimental design. **b–f** Behavioral results. **b** Discrimination between +lim and -lim is assessed at different time points after the enrichment period. Non-enriched control (NE; $n = 9$) animals do not discriminate the +lim from −lim at T1. After enrichment, the two odorants are discriminated at T1 ($n = 11$), T2 ($n = 5$), and T3 ($n = 10$) but not at T4 ($n = 9$) and T5 ($n = 5$). **c** Dec and dodec are not discriminated at any time post-learning (NE $n = 9$; T1 $n = 10$; T2 $n = 7$; T3 $n = 8$; T4 $n = 5$; T5 $n = 10$) (for clearer representation, 1 data point belonging to the Test trial of the NE group is not shown (investigation time = 6 s)). *T*-test, *$p < 0.05$. **d** Adult-born neuron density in the OB in enriched experimental groups. Groups that discriminated +lim/−lim (T1–T3) have a higher BrdU-positive cell density compared to groups that do not discriminate (T4-T5) ($n = 4$/group), t-test *$p < 0.05$. **e** Positive correlation between the discrimination index and adult-born neuron density in the OB. **f** Density of activated adult-born neurons (BrdU + /Zif268 + cells) in +lim/−lim enriched groups in response to +lim ($n = 4$/group) or dec ($n = 5$/group) and in control non-enriched in response to +lim (T1 $n = 2$; T2 $n = 4$; T3 $n = 2$; T4 $n = 3$; T5 $n = 4$). A significant increase of double-labeled cells is observed in enriched groups in response to learned odorant (+lim) in T1–T3 compared to T4-T5. Tukey tests, * at least $p < 0.05$ for comparisons between enriched animals responding to +lim. Data are represented as data points and mean ± sem. Source data are provided as a Source Data file.

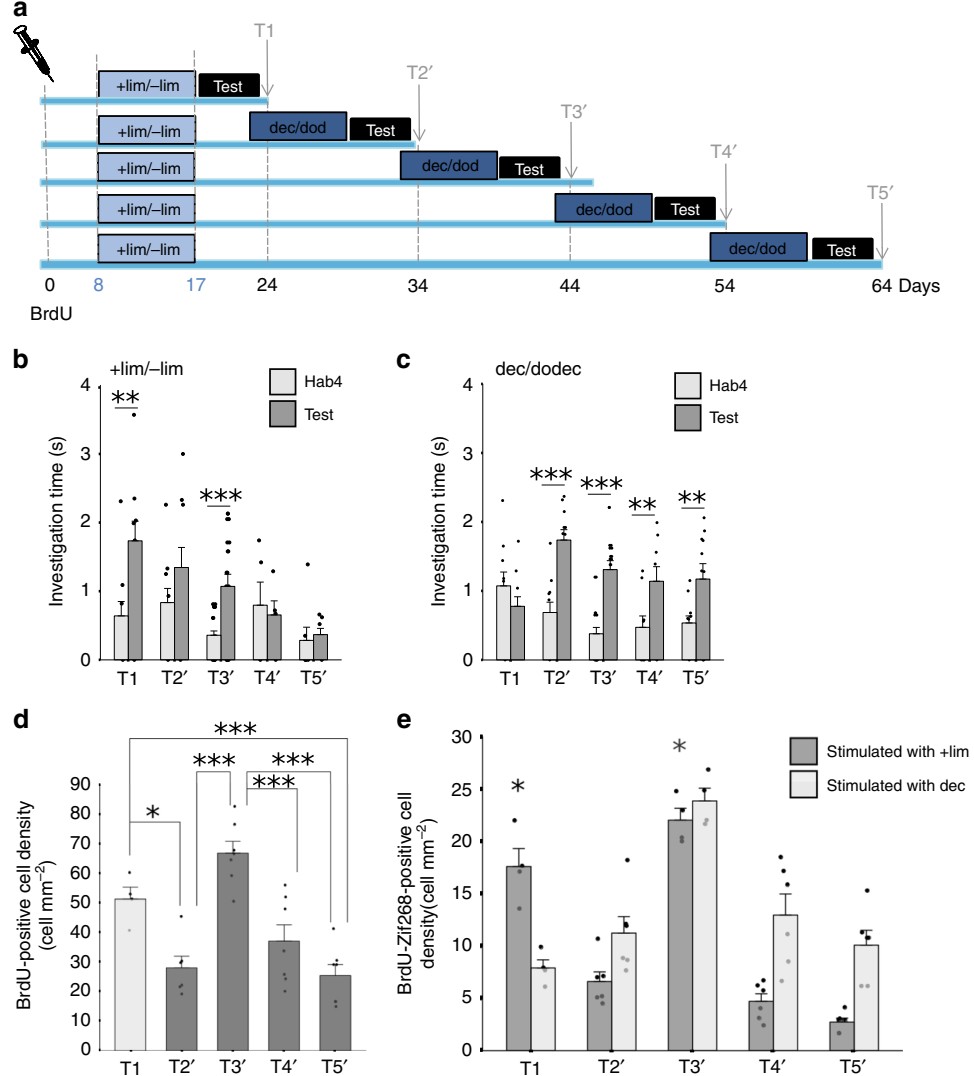

**Fig. 2 Time between learning sessions has a critical impact on memory. a** Experimental design. The groups differed by the interval between the two enrichment periods: 4, 14, 24 or 34 days separating the two enrichments. **b, c** Behavioral results. **b** Discrimination between +lim/−lim. Groups T1 and T3′ discriminate. No discrimination is observed for the other groups (T2′, T4′, T5′) (T1 $n = 11$; T2′ $n = 11$; T3′ $n = 11$; T4′ $n = 6$; T5′ $n = 7$) t-test, **$p < 0.005$; ***$p < 0.001$. **c** Dec is discriminated from dodec in Group T2′ to T5′ (T1 $n = 10$; T2′ $n = 11$; T3′ $n = 11$; T4′ $n = 10$; T5′ $n = 11$) t-test **$p < 0.005$; ***$p < 0.001$. **d** Adult-born neuron density. Higher density of adult-born neurons is observed in groups that are able to discriminate +lim from −lim (T1 and T3′) ($n = 6–7$/group), Tukey corrected t-tests *$p < 0.05$; ***$p < 0.001$. **e** Density of activated adult-born neurons (BrdU+/Zif268+ cells) in response to +lim (T1 $n = 4$; T2′ $n = 6$; T3′ $n = 4$; T4′ $n = 6$; T5′ $n = 6$) or dec (T1 $n = 4$; T2′ $n = 6$; T3′ $n = 4$; T4′ $n = 6$; T5′ $n = 6$) after successive enrichments. Density of double-labeled cells is higher in response to +lim at T1 and T3′ compared to T2′, T4′, and T5′. In response to dec, the density of BrdU/Zif268 cells is higher at T3′ compared to the other groups. T-test *$p < 0.05$ for enriched animals responding to +lim at T1 and T3′ different from all other groups. Data are represented as data points and mean ± sem. Source data are provided as a Source Data file.

odorant in groups showing the ability to discriminate +lim from −lim.

In summary, perceptual memory showed a natural memory decay between T3 and T4. This memory alteration is accompanied by a decrease in the density of adult-born neurons and in the percentage of adult-born neurons responding to the learned odorants.

**Time between learning sessions affects stored memory.** In total 4, 14, 24 or 34 days after a first enrichment with +lim/−lim, mice were enriched with a new odorant pair, dec/dodec, tested for discrimination of the two pairs of odorants and sacrificed at the same times as in the previous experiment (groups T2′ to T5′) (Fig. 2a). All groups exhibited habituation behavior (Supplementary Table 2, Supplementary Fig. 4). Regarding

discrimination, for +lim/−lim, using two-way ANOVA, we observed a group effect (T1,T2′,T3′,T4′,T5′, $F_{(4,94)} = 4.63$; $p = 0.002$) and a trial effect ($F_{(1,94)} = 19.99$; $p < 0.0001$). For dec/dodec, there was no group effect ($F_{(4,108)} = 2.19$; $p = 0.08$) but a trial effect ($F_{(1,108)} = 45.08$; $p < 0.0001$) and interaction ($F_{(4,108)} = 5.49$; $p = 0.0005$). When the second enrichment was performed 4 days after the first one (T2′), animals were only able to discriminate the second odorant pair (Supplementary Table 2, Fig. 2b, c) suggesting that the acquisition of the second task erased the memory of the first one. This result was confirmed using another strain of mice (129 mice, Supplementary Fig. 5). As an additional control, we switched the training order, the animals were enriched first with dec/dodec and then with +lim/−lim. Result was similar to that previously obtained: animals were only able to discriminate the second learned odorant pair (+lim/−lim

in this case) (Supplementary Fig. 6). When the time between the two enrichments increased (14 days, T3′), the animals were able to discriminate the two odorants of each pair (Supplementary Table 2, Fig. 2b, c) suggesting that in this experimental configuration the formation of a new memory did not impact the memory already stored. At longer times between enrichments (24 and 34 days, respectively T4′ and T5′), +lim and −lim were no longer discriminated while dec and dodec were, which is in accordance with the first experiment reporting the duration of memory retention (Supplementary Table 2, Fig. 2b, c).

BrdU-positive cell density was then assessed as in the first experiment. BrdU-positive cell density differed between groups (ANOVA, group effect, $F_{(4,26)} = 16.1$; $p < 0.0001$) (Fig. 2d). Post hoc Tukey corrected $t$-tests were then performed and showed that when the second enrichment was performed 4 days after the first one (T2′), the density of labeled adult-born cells was significantly decreased compared to T1 (Tukey corrected $t$-test $p = 0.025$). This was accompanied by impaired +lim/−lim discrimination suggesting that, at 4 days between two learning, new memory formation altered adult-born neuron survival and already stored information. However, when the second enrichment was performed 14 days after the first (T3′), both pairs of odorants were discriminated and the density of adult-born cells remained high (Tukey corrected t-test, T1 versus T3′: $p = 0.2$). Finally, when the second enrichment occurred even later (T4′ or T5′), BrdU density decreased significantly (Tukey corrected $t$-test, T3′ versus T4′: $p < 0.001$; T5′ versus T3′: $p < 0.001$) to a level comparable to T2′ (Tukey corrected $t$-test, T4′ versus T2′: $p = 0.59$; T5′ versus T2′: $p = 0.99$) which is in accordance with the first experiment (Fig. 2d). To study the involvement of adult-born neurons in processing the learned information, we assessed the density of adult-born neurons responding to +lim or to dec (group effect) at the different delays between enrichments (Fig. 2e). Using two-way ANOVA, we found a group effect ($F_{(1,42)} = 8.61$, $p = 0.0054$), a time effect ($F_{(1,42)} = 42.53$, $p < 0.0001$) and an interaction ($F_{(1,42)} = 13.08$, $p < 0.0001$). In enriched animals in response to +lim, the density of BrdU/Zif268 cells is higher at T1 and T3' compared to T2′, T4′, and T5′ (Time effect $F_{(4,21)} = 73.09$, $p < 0.0001$; Tukey corrected $t$-test $p < 0.05$) which is in accordance with the performance of +lim versus −lim discrimination. In enriched animals in response to dec, the density of BrdU/Zif268 cells is higher at T3′ compared to the other groups (Time effect $F_{(4,21)} = 12.08$, $p < 0.0001$; Tukey corrected $t$-test $p < 0.05$). These changes in density relied at least partly on changes in percentage of BrdU/Zif268 double-labeled cells (Supplementary Fig. 7). For the T2′, the decrease in the BrdU-Zif268 density is due more to a decrease in BrdU-positive cell density than to a decrease in the percentage of BrdU/Zif268 double-labeled cells. However, overall, the percentage of BrdU/Zif268-positive cells was higher in groups that discriminate +lim from −lim (T1 and T3′) than in groups that do not discriminate (T2′, T4′, and T5′) ($p = 0.007$).

In summary, the time between the two enrichments is critical to the upkeep of the memory already stored and associated adult-born neurons survival and functional involvement. These data further suggest that the survival of at least some of these adult-born neurons is not assured as, for one week after learning, they go through a fragile state, which is vulnerable to a new learning challenge.

**Sensory reactivation prevents cell and memory loss.** With a short period between the two learning sessions (T2′), we found that the new memory altered previously stored information and induced apoptosis of adult-born neurons. Since cell survival is known to be input-dependent, we tested whether reactivating the network that responded to the odorants used for the enrichment

would avoid this overwriting of the memory. To do this we maintained exposure to the first pair of odorants during enrichment with the second pair (Fig. 3a).

In all groups, mice exhibited habituation behavior (Supplementary Table 3, Supplementary Fig. 8). Using two-way ANOVA, we observed significant effects of trial and group (trial: $F_{(1,164)} = 16.88$; $p < 0.0001$, group: $F_{(3,164)} = 12.55$; $p < 0.0001$) (Fig. 3b) for +lim/−lim discrimination and a trial effect and a group effect (trial: $F_{(1,164)} = 31.33$; $p < 0.0001$, group: $F_{(3,164)} = 3.14$; $p = 0.026$) (Fig. 3c) for dec/dodec discrimination. As in the first experiment (Fig. 1a), we observed that +lim/−lim enrichment improved discrimination between these two odorants when tested at T2' (group 1, see Supplementary Table 3, Fig. 3b, c) and that the second enrichment with dec/dodec when performed 4 days after the first one erased the ability to discriminate +lim/−lim (group 2, Fig. 3b, c, Supplementary Table 3). Interestingly, if we maintained the enrichment with +lim/−lim during the second enrichment period (group 3), the animals were able to discriminate both odorant pairs (Fig. 3b, c, Supplementary Table 3). This prevention of forgetting can be due to network reactivation or relearning. Finally, when the retention of the task was tested at T3′, mice were, as previously, able to discriminate both odorant pairs (group 4, Fig. 3b, c, Supplementary Table 3).

To understand the dynamics of learning-dependent survival of adult-born neurons underlying these effects, we tagged two adult-born neuron populations differing in age using two analogues of thymidine, ChlorodeoxyUridine (CldU) and IododeoxyUridine (IdU). More precisely, we labeled a first pool of 8-day-old adult-born cells in the OB at the beginning of the first enrichment (with +lim/−lim) with CldU and a second pool of 8-day-old adult-born cells in the OB at the beginning of the second enrichment (with dec/dodec) but not present in the OB at the beginning of the first enrichment (Fig. 3a, d) with IdU. The densities of CldU- and IdU-positive cells were assessed in the OB at T2. Overlapping between CldU and IdU was never observed. Results showed differences between experimental groups for both markers (ANOVA, CldU: $F_{(3,16)} = 19.06$, $p < 0.0001$; IdU: $F_{(3,24)} = 7.8$, $p = 0.0008$). Post hoc Tukey corrected $t$-tests revealed that the level of CldU-positive cells was lower in group 2 compared to the other groups, in accordance with +lim/−lim discrimination performance (group 2 versus group 1: $p = 0.056$, group 2 versus group 3: $p = 0.0029$, group 2 versus group 4: $p < 0.001$; Fig. 3e). Interestingly, the density of CldU-positive cells was not significantly higher in group 3 compared to group 1 ($p = 0.48$) suggesting that the second enrichment recruited a new pool of adult-born neurons. Supporting this hypothesis, all groups that were submitted to dec/dodec enrichment after the +lim/−lim enrichment, including group 2, showed a higher level of IdU-positive cells than group 1 (respectively $p = 0.03$, $p = 0.017$ and $p < 0.001$) (Fig. 3f). This strongly suggests that the second learning recruited a new population of adult-born neurons younger than that recruited during the first learning.

We learned from this experiment that reactivating the network of the first learned odorant pair during the second task prevented memory erasure and the disappearance of adult-born cells. Thus, for a short time between two successive learning tasks, adult-born neurons can be maintained in the network providing that the relevant sensory inputs remain in the environment. However, despite the presence of these adult-born neurons still in their critical period of integration, a distinct pool of younger adult-born neurons was recruited by the second task. It remains the question of whether the second learning relies on the last recruited adult-born neuron population (IdU-positive cells) or on both adult-born neuron populations (IdU- and CldU-positive cells). In other words, are adult-born neurons allocated to a memory trace able to underlie or contribute to subsequent learning?

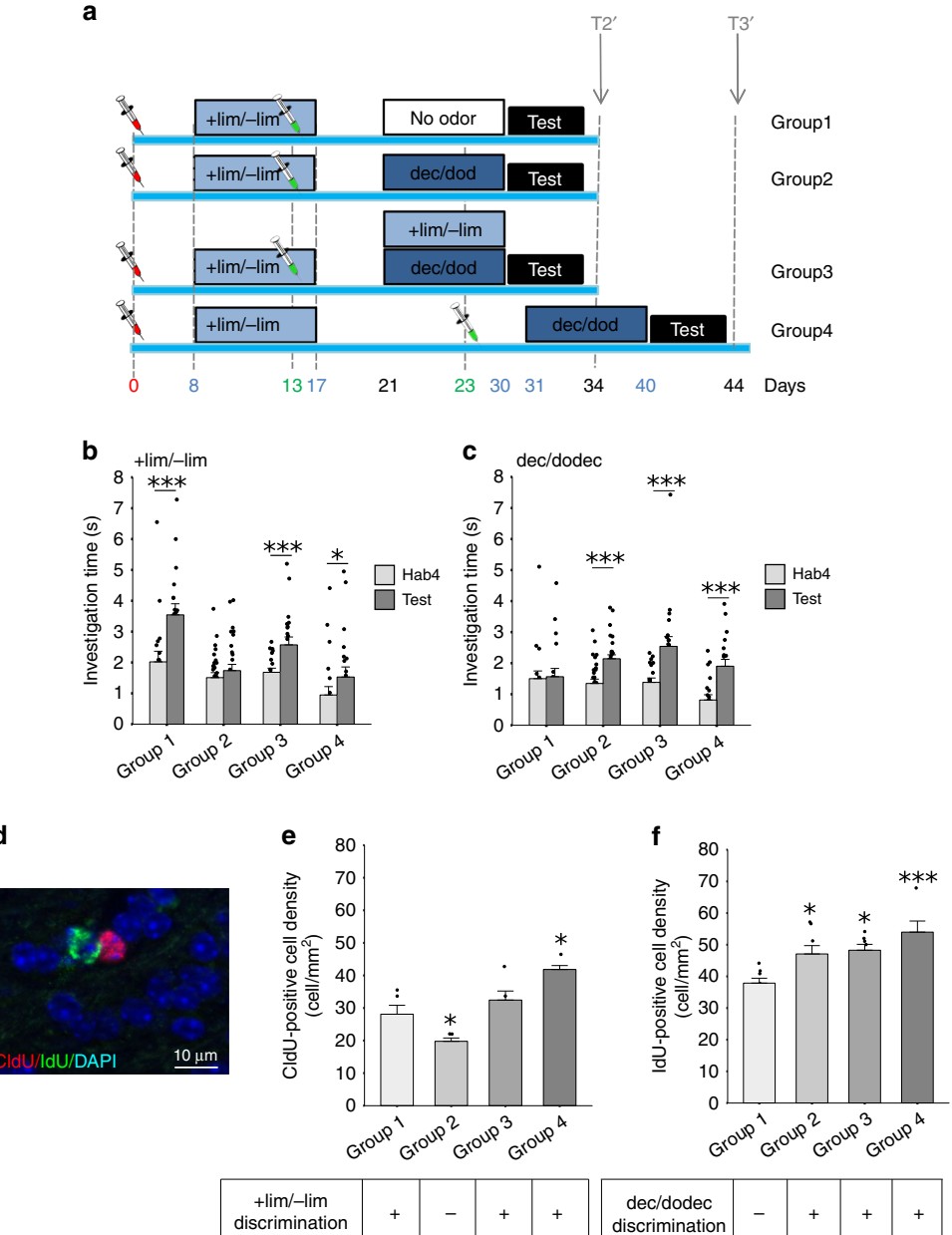

**Fig. 3 Reactivation of learned information prevented cell death and maintained memory. a** Experimental design. **b**, **c** Behavioral results. **b** +lim and -lim are discriminated in groups 1 ($n = 18$), 3 ($n = 20$) and 4 ($n = 10$) but not in group 2 ($n = 18$). **c** Dec/dodec are discriminated in Group 2 ($n = 18$), 3 ($n = 20$) and 4 ($n = 10$) but not in group 1 ($n = 18$). **d** Example of CldU/DAPI and IdU/DAPI double labeling. **e**. Density of CldU-positive cells. A lower density is observed in group 2 compared to groups 1, 3, and 4 and a higher density in group 4 compared to groups 1, 2, and 3 ($n = 5$/group). **f** Density of IdU-positive cells is higher in groups 2 ($n = 8$), 3 ($n = 7$) and 4 ($n = 5$) compared to group 1 ($n = 8$). Tukey corrected $t$-tests *$p \leq 0.05$, ***$p < 0.001$. Data are represented as data points and mean ± sem. Source data are provided as a Source Data file.

**Distinct adult-born neuron cohorts for distinct memories.** We used the same learning paradigm as previously (group 3; Figs. 3a and 4a) in which mice underwent a first enrichment period with +lim/−lim and 4 days later a second one with dec/dodec in the presence of +lim/−lim to avoid memory loss. Eight days before the first enrichment period, adult-born neuron progenitors were transduced with halorhodopsin-expressing or control lentiviruses coupled with EYFP expression and animals were implanted with optical fibers in the OB. Light stimulation was used to block adult-born cell activity during the test trial of the habituation/dishabituation task. Results showed that light-triggered inhibition of neurons born 8 days before the first learning task altered the discrimination of +lim/−lim (Fig. 4b, Supplementary Table 4)

but not of dec/dodec (Fig. 4c, Supplementary Table 4). Animals infected with a control virus and submitted to light stimulation behaved as group 3 of the previous experiment, suggesting no direct effect of the light. To test that light-triggered inhibition of adult-born neurons in halorhodopsin animals had no deleterious effect on spontaneous odor discrimination, we assessed the discrimination of a pair of dissimilar odorants (+lim/+carvone) and found that it was not impaired by light (Fig. 4d, Supplementary Table 4). We then analyzed the level of viral transfection in control and halorhodopsin groups and found no difference of EYFP-positive cell density ($t$-test, $p = 0.5$, Fig. 4e). We also verified the effectiveness of light-mediated inhibition of adult-born granule cells by assessing the expression of Zif268-positive cells in

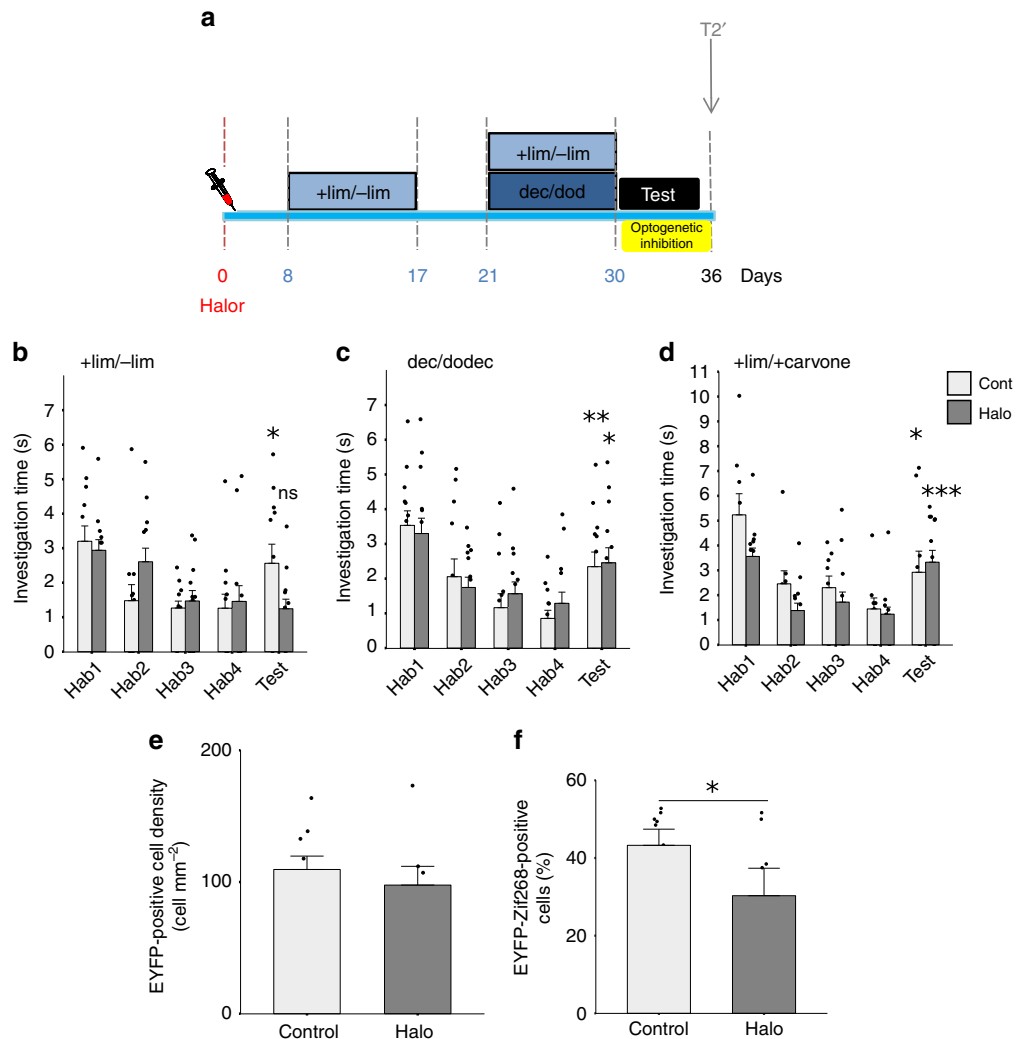

**Fig. 4 Optogenetic inhibition revealed that distinct adult-born neuron cohorts underlined distinct memories. a** Experimental design. Adult-born neurons aged 8 days at the beginning of the first enrichment were transduced with a lentivirus expressing halorhodopsin (Halo) or with an empty virus (Cont). Light stimulation was performed during the test trial of the habituation/dishabituation task. **b–d** Behavioral results. **b** The learned discrimination of +lim/−lim is abolished by light stimulation in the Halo group ($n = 13$), but not in the Control group ($n = 12$). **c** Discrimination between dec/dodec is not altered by light stimulation (Cont $n = 13$; Halo $n = 14$). **d** Discrimination between a dissimilar odorant pair +lim/ +carvone is not altered by light-triggered inhibition of adult-born neurons (Cont $n = 9$; Halo $n = 14$). $t$-tests *$p < 0.05$, **$p < 0.01$, ***$p < 0.001$ (for difference between Hab4 and Test in **b–d**). **e** The density of EYFP-positive transduced cells in the OB is similar between Control ($n = 9$) and Halo ($n = 7$) animals. **f** Percentage of EYFP/Zif268-positive cells after light-triggered inhibition is decreased in Halo ($n = 7$) compared to Control ($n = 8$). $t$-test *$p < 0.05$ (for difference between Halo and Control groups in **f**). Data are represented as data points and mean ± sem. Source data are provided as a Source Data file.

EYFP-positive neurons after stimulation with light. The expression of Zif268 was significantly lower in the halorhodospin compared to the control group ($t$-test, $p = 0.03$, Fig. 4f).

Hence, silencing a cohort of adult-born neurons aged between 8 and 18 days at the time of learning leads to memory erasure demonstrating the crucial role of adult-born neurons in perceptual memory. However, this manipulation did not affect the memory of a new discrimination task learned shortly afterwards, suggesting that these adult-born neurons are allocated to only one memory trace.

## Discussion
The OB is continuously receiving waves of adult-born neurons and so contains adult-born neurons of different ages and levels of maturation[18]. When learning happens successively, which neurons take part in underlying behavioral changes? When the time between the two learning tasks is short (4 days), part of the adult-

born neurons present in the OB after the first task are prematurely killed and memory is lost. This conclusion is based on the loss of BrdU-positive neurons and is consistent with previous reports showing that blocking adult-born cell death prevented memory erasure[19]. In addition to training-dependent changes in BrdU-positive cell density, the functional involvement of these cells in response to the learned odorant increased with performance for neurons aged 8 days at the beginning of learning. We further show that these two learning tasks, despite they are spaced by a short 4-day time period, induced two distinct waves of adult-born neuron survival and that optogenetically inactivating the first cohort altered the first memory, without affecting the second learning. This further reinforces the idea that the 20-day old BrdU-tagged neurons exited the time window during which they are most crucial to enrichment-induced improvement in discrimination. It is worth noting that in the short time condition, the number of adult-born cells is reduced but a substantial fraction of adult-born neurons (>20%) still respond to the learned odorant even though

discrimination is lost. This state differs from that observed for longer between-task times where both density of adult-born neurons and their responsiveness to the learned odorants are strongly reduced. This suggests only a partial withdrawal of the remaining adult-born neurons from processing the first learned odorants in the short time condition, sufficient however to prevent discrimination. Nevertheless, we conclude that cohorts of neurons of different ages, but still within their critical period, are required to allow the encoding of successive learning experiences. Importantly, when the time between tasks is longer (14 days), no cell loss occurred and performances were maintained. Altogether, these data provide evidence for an unexpectedly early and sharp functional transition within the critical period. The critical period of adult-born neurons which, based on their vulnerability to cell death, morphological development and synaptic current development and plasticity, has been reported to last from 30 days to 60 days post neuronal birth[6,10,20–23]. Within this critical period, our data suggest that availability of adult-born neurons for encoding new experience lasts no longer than 20 days.

When previously learned information is reintroduced into the environment, we observed no premature cell death and the second learning did no longer erase the first one. To maintain the first learned discrimination, the network could either use adult-born neurons already integrated into this network or recruit a new cohort of adult-born neurons. Optogenetic inactivation of the 8-day old cohort of adult-born neurons present in the OB at the beginning of the first training impaired memory of the first task. This strongly suggests that performance relies on adult-born neurons recruited in the initial task suggesting that the memory is recalled rather than re-encoded. Hence, reactivated memories are not over-written and forgotten. On the contrary, neurons encoding obsolete information (i.e. not present in the environment) could be sentenced to death after about 45 days unless another learning task occurs soon after, causing interference and leading to earlier neuronal death. Our data thus suggest that adult-born neurons are sensitive to interference only during their first 3 weeks and that if further learning occurs when adult-born neurons present in the OB are more than 21 days old, they are resistant to death and memory persists. On the other hand, a second learning task causes neurons aged less than 21 days old to die and erases the encoded memory. This difference could be due to the state of synaptic integration of the adult-born neurons[20] which are more or less sensitive to competition from later waves of adult-born neurons.

Much of what we experience is ultimately forgotten, but memories for some events persist. Here, we report that modulation of the OB circuitry, as a function of the environment and dependent on the fate of maturing adult-born neurons, is responsible for the balance between the transience or persistence of memory. This is made possible thanks to distinct neuronal populations encoding temporally distinct experiences.

## Methods

**Mice**. Adult C57BL/6J mice (8 weeks old, male, Charles River, L'arbresles, France) and 129 mice (8 weeks old, male, Charles River, L'arbresles, France) were used in this study. They were housed in standard laboratory cages with water and food ad libitum and were kept on a 12-hr light/dark cycle at a constant temperature of 22 °C. All behavioral training was conducted in the afternoon (12:00–18:00). Experiments were done following procedures in accordance with the European Community Council Directive of 22 September 2010 (2010/63/UE) and approved by the National Ethics Committee (Agreement DR2013-48(vM)). Every effort was made to minimize animal suffering.

Different animals were used in different experimental groups. Histological data were obtained from a sample of animals used for behavioral tasks.

**Experimental designs**. Regarding the first experiment, mice were injected with a DNA marker, BrdU, in order to label a cohort of adult-born neurons. Eight days later, they began the enrichment procedure with (+)limonene (+lim) and (−)limonene (−lim) for one hour daily over 10 days. At the end of the enrichment, mice were

tested on their spontaneous discrimination between +lim and −lim and also between another pair of perceptually similar odorants (decanal (dec) and dodecanone (dodec)). Discrimination was assessed using an olfactory habituation/dishabituation task. Animals were sacrificed 24, 34, 44, 54 or 64 days after BrdU injections (Fig. 1a).

Regarding the second experiment, eight days after BrdU injection, a different set of mice was similarly enriched with +lim and −lim but this was followed by a second 10-day enrichment period with dec and dodec either 4, 14, 24, 34 or 44 days after the first. At the end of both enrichments, mice were tested on their spontaneous discrimination between the two odorants of each of the pairs using an olfactory habituation/dishabituation task and animals were sacrificed 24, 34, 44, 54 and 64 days post BrdU injections (Fig. 2a)

In the third experiment, we used two other DNA markers (analogues of BrdU), 5′-chloro-2′-deoxyuridine (CldU) and 5′-iodo-2′-deoxyuridine (IdU), to label two different populations of adult-born cells. CldU was injected 8 days before the first enrichment (again with +lim and -lim) while IdU was injected 8 days before the second enrichment period. The second enrichment period varied among groups: no enrichment (group 1); dec and dodec (group 2 and group 4); +lim and −lim plus dec and dodec (group 3). The time between the two enrichments was either 4 or 14 days. As previously, discrimination was assessed using a habituation/dishabituation task and animals were sacrificed 34 or 44 days post-CldU injection (Fig. 3a).

In the fourth experiment using group 3 experimental configuration from the previous experiment, we performed targeted lentiviral-induced halorhodospin channel expression (NpHR3.0) in the subventricular zone, and used optogenetics to specifically inhibit the population of adult-born neurons arriving in the OB at the beginning of the +lim/−lim enrichment (Fig. 4a).

**Perceptual learning**. This implicit olfactory learning consisted in passive exposure to odorants (enrichment). For the olfactory enrichment, 100 μL of pure odorant were placed in two separate tea balls of the home cages for one hour daily over 10 days. For the multiple enrichment (experiment 3, group 3), the two pairs of odorants were presented with an interval of 1 h (SI Methods).

We assessed the spontaneous discrimination between two pairs of similar odorants: +lim/−lim and dec/dodec using olfactory habituation/dishabituation. A test session consisted of four 50-s odor presentations of a first odorant (Hab) at 5 min intervals, followed by one 50-s presentation of the second odorant of the pair (Test). Investigation was defined as active sniffing within 1 cm of the tea ball (SI Methods)[8,24,25]. Exploration assessments were done blind with regard to the experimental group.

**Data analysis**. Data analysis was performed using R software (CRAN). Normality was assessed using the Kolmogorov-Smirnov test. Global two-way ANOVAs were performed to evaluate changes in discrimination abilities between groups. Then intra-group one-way RM-ANOVAs and unilateral paired t-tests were performed to determine whether the mice exhibited habituation (trial effect) as well as discrimination (by comparing Hab4 and Test). Discrimination was indicated by a significant increase in investigation time during the test trial. Discrimination index was calculated as [1-(Hab4/Test)][26] (SI Methods). Sample sizes were determined based on previous reports[8,14,27]. Animal assignation to the various experimental groups was randomized.

**Adult-born cell density assessment**. Bromodeoxyuridine (BrdU) (Sigma-Aldrich) was injected 8 days before the enrichment period (50 mg kg$^{-1}$ in saline, 3x at 2-h intervals; i.p.)[14,22].

24, 34, 44, 54 and 64 days post BrdU injection, five mice were taken randomly from each experimental group for sacrifice and BrdU, IdU and CldU immunohistochemistries were carried on as described in SI Methods[14,22].

The method used for BrdU-, IdU- and CldU-positive cell counting is described in SI Methods[14,28]. The mean positive cell density of each array was calculated and averaged within each experimental group. Between-groups comparisons of the mean cell density were performed by ANOVA followed by post hoc t-tests with Tukey corrections. Unilateral t-tests were performed for comparisons between two groups. The level of significance was set to 0.05.

**BrdU/Zif268 experiment**. To investigate immediate early gene expression in response to odorant exposure, mice were presented with a tea ball containing 100 μl of pure +lim or dec for 1 h, 1 h before sacrifice. A rabbit anti-Zif268 (1:1000, Santa Cruz Biotechnology) and a rat anti-BrdU (1:100, Harlan Sera-Lab) were used. The appropriate secondary antibodies, coupled to Alexa Fluor 633 and 488 (Invitrogen) were used for revelation of the different markers. BrdU/Zif268 density was calculated by combining the mean percentage of BrdU/Zif268 positive cells per group to individual density of BrdU positive cells. All cell counts were conducted blind with regard to the experimental group.

**Optogenetics in freely behaving mice**. Hundred and fifty nanoliters of pLenti-hSyn-eNpHR3.0-EYFP lentivirus (9.22 × 10$^6$ IU ml$^{-1}$) or 300 nL of control pLenti-hSyn-EYFP (1.1 × 10$^6$ IU ml$^{-1}$, expressing only the reporter gene EYFP[29]) injections were done bilaterally in the subventricular zone, with the following coordinates respective to bregma: antero-posterior + 1 mm, medio-lateral ± 1 mm, dorso-ventral −2.3 mm and at a rate of 150 nL min$^{-1}$. Just after virus infusions, mice were

implanted with bilateral optic fibers (200 nm core diameter, 0.22 N.A., Doric Lenses) in the OB, with the following coordinates respective to bregma: antero-posterior +4.6 mm, medio-lateral ±0.75 mm, dorso-ventral −2 mm. Mice were injected with a ketoprophen solution (2 mg kg$^{-1}$) after the surgery as well as during the following days and allowed to recover with food and water ad libidum.

During the habituation/dishabituation mice were automatically stimulated (crystal laser, 561 nm, 10–15 mW, continuous stimulation) only during the test trial (Test) when they entered a 2.5 cm diameter zone around the tea ball.

In order to control of light-triggered inhibition, 36 days post-surgery and lentiviral infusion, mice were stimulated during the hour before sacrifice with light patterns mimicking the average light stimulating pattern during the test trial (0.75 s light ON, 5 s light OFF for 1 h). After brain sectioning (see above), EYFP and Zif268 double immunohistochemistry was performed: incubation with rabbit Zif268 antibody (1:1.000, Santa Cruz, ref: Sc-189), chicken GFP antibody (1:1.000, Anaspec TEBU, ref. [29], 55423). The density of EYFP, Zif268$^+$ and double-stained cells were counted on 1–2 slices under the fiber implantation to allow assessment of the inhibition. Statistical significance was assessed using a unilateral t-test.

**Reporting summary**. Further information on research design is available in the Nature Research Reporting Summary linked to this article.

## Data availability
The data associated to all main figures are provided as a Source Data file.

## Code availability
The codes are provided in supplementary file entitled Supplementary Software 1.

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

## Acknowledgements
This work was supported by CNRS, Inserm, Lyon 1 University and Ecole Normale Supérieure de Lyon. We would like to thank C. Benetollo from the Neurogenetic and Optogenetic Platform of the CRNL for lentiviral production and G. Froment, D. Nègre and C. Costa from the lentivector production facility /SFR BioSciences de Lyon (UMS3444/US8).

## Author contributions
J.F., M.M., A.D., and N.M. designed the research. J.F., M.M., A.Z, M.C., J.S., and L.C. performed the research. J.F., M.M., M.R., A.D., and N.M. wrote the paper.

## Competing interests
The authors declare no competing interests.
