## [Peer Review File · Nature Communications]

Reviewers' Comments:

Reviewer #1:

Remarks to the Author:

The manuscript by Forest et al. described a cluster of interesting findings in olfaction memory. They designed some sophisticated experiments using two successive olfactory discrimination tasks. They found that the time between the two tasks matters for a batch of newborn neurons likely involved in memory coding (the first task). Lastly, they used optogenetic stimulation showing that activation of this population of neurons affected the first task with no obvious on the second task. The work is interesting, addressing an important question in olfactory neurogenesis. Overall, most conclusions are convincing, but we do have several major concerns as below.

Major concerns:

1) All the conclusions have been drawn from correlations between BrdU+ cells and behavioral outcome. In either Figure 2 or 3, it will be necessary for the authors to provide another set of data showing whether the behavior outcome is related to the number of BrdU+ and an activity marker+ cells. If the correlation is weak, the significance of current conclusion will become weak.

2) In figure 2, the authors found that dec/dod training in a short time window caused loss of discrimination that previously learned. Interestingly or surprisingly, they found a decrease of BrdU+ cells. WHY? How could this happen? Will a higher ratio of cell death occur? Does it mean that the dec/dodec training facilitates the extinction to enhance cell death? In figure 3 group 3, if the training order of +lim/-lim and dec/dod is changed, what will happen? If use the dec/dod as the first training and the +lim/-lim as a second, will the authors see the same decrease in T2?

3) In figure 1, The NE group data should be moved into an independent plot. You can compare it with T1, but no comparison with T2-T5. Therefore, in the same plot it causes a feeling to readers that you have done controls for T2-T5. Following this concern, it will be extremely informative if the authors could provide a declining number of BrdU+ cells for NE analyzed at the time points of T2-T5. Will the +/-lim training facilitate/dampen the decline of the number of BrdU+ cells across the T2-T5.

Minor concerns:

1) The descriptions of most experiments are short of details. For example, how was the training of +lim/-lim performed?...

2) It is hard to conclude one neuron-for-one memory principle, although it is reasonable to conclude short-term availability.

Reviewer #2:

Remarks to the Author:

This manuscript by Forest and colleagues explores the encoding of perceptual olfactory discrimination by adult olfactory newborn cell populations. They use a combination of behavior, cell proliferation markers, immunohistochemistry and optogenetics to determine the contribution of select adult born olfactory granule cell populations (based on their age) to the expression of olfactory discrimination to two separate pairs of odors. They first establish that mice lose the ability to distinguish between two previously exposed (enriched) odors between 27 and 37 days after removal of the odors. Loss of this discrimination ability correlated with a decrease in BrdU cell density in the granule cell layer. To

explore the role of interference, and the potential contribution of different neuronal populations to the discrimination of multiple olfactory odor pairs, the authors added exposure to a second odor pair. They found that addition of a second odor pair within 4 days of the end of exposure to the first odor pair impedes discrimination of the first pair, and decreases BrdU density. This is prevented if exposure to first pair is concurrent with that of the second pair. Use of two proliferation markers, CldU and IdU, also showed that a second cohort of neurons, which were at least 8 days old at the second exposure, did not show a decrease in numbers. To establish whether separate neuronal populations are responsible for representing the two odor pairs, the authors conduct an optogenetics experiment. They infused halorhodopsin-expressing lentivirus into the subventricular zone 8 days prior to the start of the first odor enrichment, and silenced them during test. They found that this impaired discrimination of the first pair, but not the second, suggesting a separate population of neurons is involved in the discrimination of the second pair.

These data investigate a very fundamental question, that of how multiple representations and memories are processed in the brain. Whereas some progress has been made in this regard in other systems (amygdala, hippocampus), much less is known about this regulation in the olfactory system. Adult neurogenesis provides an excellent opportunity to ask these important questions. While the questions and data raised by the authors are important and of interest, additional experiments would help strengthen their conclusions.

Major Concerns:

- The cell counts for BrdU, CldU and IdU are extremely low ($\sim 3 \text{ cells}/\mu\text{m}^2 \times 10^{-5}$). This is not a standard unit for this. If this is correct, it is hard to ascertain meaning to such small values. The manuscript would benefit from multiple injections or a transgenic strategy to increase their tagging efficiency.
- In figure 3, the authors argue that a decrease in CldU numbers in group 2 matches the decrease in performance. However, there is no significant difference between group 2 and group 1, which performs the discrimination, weakening their claim. In addition, these are not cells activated by exposure to the odor, but cells positive for the proliferation marker. Similarly, the authors argue that "adult-born neurons are sensitive to interference only during their first 3 weeks and that if further learning occurs when adult-born neurons saved by the first are more than 21 days old, they are resistant to death and memory persists.". However, they do not tag the neurons to demonstrate it is indeed the neurons involved in the first learning that are dying. To more strongly demonstrate this, an activity dependent tagging approach is necessary, particularly given the previous point about cell numbers.
- This perceptual olfactory discrimination protocol has been published several times and very successfully by this group. However, replication in a mouse strain that considerably explores odors (129/C57?), in which exploration levels - particularly between Hab4 and test - would be greater than 1 second, would make these findings significantly more convincing from a behavioral perspective.

Minor points

- Is there any overlap between CldU and IdU positive cells?
- Figure 4D shows a very small reduction in cell activity with their optogenetic manipulation (about 10% of cells). This needs to be justified by the authors, especially if their cell density is as low as shown in Figure 4C.

Point-by-point responses to referees' comments

Reviewer #1 (Remarks to the Author):

The manuscript by Forest et al. described a cluster of interesting findings in olfaction memory. They designed some sophisticated experiments using two successive olfactory discrimination tasks. They found that the time between the two tasks matters for a batch of newborn neurons likely involved in memory coding (the first task). Lastly, they used optogenetic stimulation showing that activation of this population of neurons affected the first task with no obvious on the second task. The work is interesting, addressing an important question in olfactory neurogenesis. Overall, most conclusions are convincing, but we do have several major concerns as below.

Major concerns:

1) All the conclusions have been drawn from correlations between BrdU+ cells and behavioral outcome. In either Figure 2 or 3, it will be necessary for the authors to provide another set of data showing whether the behavior outcome is related to the number of BrdU+ and an activity marker+ cells. If the correlation is weak, the significance of current conclusion will become weak.

We added new Figures 1D and 2D and two new supplementary figures in which we present the density and percentage of adult-born neurons expressing Zif268 in response to learned and unlearned odorants in the different conditions.

In Figure 1D, BrdU/Zif268 data are consistent with BrdU data and suggest that modulation in the density of BrdU-positive cells parallels the modulation in their functional involvement, reinforcing the significance of the role of adult-born neurons in olfactory memory coding.

In Figure 2D, the density of BrdU/Zif268 + adult born cells aged 8 days at the time of the first learning, in response to the learned odorants also closely reflects –lim/–lim discrimination ability over time. Regarding the response to dec, used in the second learning, the density of BrdU/Zif268 + cells is not fully correlated to olfactory performances, possibly due to the age of BrdU+ cells at the time of the second learning. Indeed, these neurons are at least 20 day-old at the time of the second learning. Considering that their optogenetic inactivation does not impair discrimination performances of dec/dodec, this further reinforces the idea that the 20-day old BrdU-tagged neurons exited the time window during which they are most crucial to enrichment-induced improvement in discrimination. These new elements are discussed in the discussion section.

2) In figure 2, the authors found that dec/dod training in a short time window caused loss of discrimination that previously learned. Interestingly or surprisingly, they found a decrease of BrdU+ cells. WHY? How could this happen? Will a higher ratio of cell death occur? Does it mean that the dec/dodec training facilitates the extinction to enhance cell death?

To answer this puzzling issue, we would like first to discuss the literature and second to present an additional experiment that we performed.

The BrdU experiments, with injections of the marker before any manipulation, are routinely used to assess survival of adult-born neurons. Indeed, less BrdU positive cells retrieved is an indicator of less survival and thus of cell death. We chose this strategy to assess cell death over the entire training period. This is the reason why we conclude the dec/dodec training in a short time window after +lim/–lim training caused a loss of adult-born cells.

Nevertheless, we tried to detect directly adult-born cell apoptosis in the olfactory bulb by immunocytochemistry of the activated Caspase 3. We set up 2 new experimental groups. A group was submitted to the two learning tasks within the short delay (T2') but the second learning was interrupted after 2 or 4 days (see below, group 2, delays d2 and d4, pooled in the analysis) in order to

capture cell death during the time window it may occur. This delay was chosen based on a previous work using associative olfactory and non-olfactory training in which we showed that BrdU-positive cell loss can occur rather quickly during extinction (Sultan et al J Neurosci 2011). Group 1 was a control non-enriched group sacrificed at the same delays.

Results are shown below.

We faced the strong limitation of this approach. In contrast to the cumulative BrdU approach, it captures only a fraction of the cells dying at the time of animal's death, leading to an underestimate of total cell death. This is due to the presumed short time window of caspase 3 activation during the process of apoptosis and to the fact that apoptosis is not synchronized in the tissue. As a result, we were able to retrieve only a small number of activated caspase3-positive cells, making difficult any statistical analysis, even though a trend toward more apoptotic cells was revealed in Group 2, as expected.

Regarding the important issue raised by the reviewer about the link between extinction and adult-born cell death, we cannot conclude from the current experiment, but in a previous experiment using another type of learning, we showed that blocking cell death prevented behavioral extinction, suggesting that adult-born neurons are required to support memory (Sultan et al J neurosci 2011). This reference is now added to the discussion.

In figure 3 group 3, if the training order of +lim/-lim and dec/dod is changed, what will happen? If use the dec/dod as the first training and the +lim/-lim as a second, will the authors see the same decrease in T2?

We performed the experiment suggested by the reviewer (see new Supplementary Figure 6) and found that the training order does not affect the outcome.

3) In figure 1, The NE group data should be moved into an independent plot. You can compare it with T1, but no comparison with T2-T5. Therefore, in the same plot it causes a feeling to readers that you have done controls for T2-T5. Following this concern, it will be extremely informative if the authors could provide a declining number of BrdU+ cells for NE analyzed at the time points of T2-T5. Will the +/-lim training facilitate/dampen the decline of the number of BrdU+ cells across the T2-T5.

We now have included non-enriched groups at all time points and moved these data to a new figure (Supplementary Figure 2).

No time effect was observed in non-enriched animals ($F(4,24)=0,32$ $p=0.85$). This is consistent with both our work (Mandaïron et al Neurosci 2006) and Dr Mori's group data (Yamaguchi and Mori PNAS 2005,) showing no difference in BrdU positive cells between respectively 15 and 60 days or between 28 and 60 days post birth of neurons. These data are less consistent with Petreanu et al (JN 2002) using tritiated thymidine and showing a decline between 15 and 45 days. Difference in the marker used as well as the mice strain (CD1 versus C57Bl) could at least partly explain these differences. In addition, Yamaguchi and Mori (2005) as well as Petreanu et al (2002) showed an increased in BrdU positive cells between 7 and 14 days post proliferation marker injection, followed by quick decline between 14 and 21 days. This increase occurs within the time window of the first learning task in our study and may correspond to the BrdU-positive cell surviving in trained animals and not in control animals.

Minor concerns:

1) The descriptions of most experiments are short of details. For example, how was the training of +lim/-lim performed?...

In methods and Supplementary methods. training procedure for +lim/-lim has been clarified.

2) It is hard to conclude one neuron-for-one memory principle, although it is reasonable to conclude short-term availability.

The manuscript title was rewritten according to the reviewer's suggestion: " Short term availability of adult-born neurons for memory encoding".

Reviewer #2 (Remarks to the Author):

This manuscript by Forest and colleagues explores the encoding of perceptual olfactory discrimination by adult olfactory newborn cell populations. They use a combination of behavior, cell proliferation markers, immunohistochemistry and optogenetics to determine the contribution of select adult born olfactory granule cell populations (based on their age) to the expression of olfactory discrimination to two separate pairs of odors. They first establish that mice lose the ability to distinguish between two previously exposed (enriched) odors between 27 and 37 days after removal of the odors. Loss of this discrimination ability correlated with a decrease in BrdU cell density in the granule cell layer. To explore the role of interference, and the potential contribution of different neuronal populations to the discrimination of multiple olfactory odor pairs, the authors added exposure to a second odor pair. They found that addition of a second odor pair within 4 days of the end of exposure to the first odor pair impedes discrimination of the first pair, and decreases BrdU density. This is prevented if exposure to first pair is concurrent with that of the second pair. Use of two proliferation markers, CldU and IdU, also showed that a second cohort of neurons, which were at least 8 days old at the second exposure, did not show a decrease in numbers. To establish whether separate neuronal populations are responsible for representing the two odor pairs, the authors

conduct an optogenetics experiment. They infused halorhodopsin-expressing lentivirus into the subventricular zone 8 days prior to the start of the first odor enrichment, and silenced them during test. They found that this impaired discrimination of the first pair, but not the second, suggesting a separate population of neurons is involved in the discrimination of the second pair.

These data investigate a very fundamental question, that of how multiple representations and memories are processed in the brain. Whereas some progress has been made in this regard in other systems (amygdala, hippocampus), much less is known about this regulation in the olfactory system. Adult neurogenesis provides an excellent opportunity to ask these important questions. While the questions and data raised by the authors are important and of interest, additional experiments would help strengthen their conclusions.

We thank the reviewer for these positive comments and for your suggestions.

Major Concerns:

- The cell counts for BrdU, CldU and IdU are extremely low ($\sim 3 \text{ cells}/\mu\text{m}^2 \cdot 10^{-5}$). This is not a standard unit for this. If this is correct, it is hard to ascertain meaning to such small values. The manuscript would benefit from multiple injections or a transgenic strategy to increase their tagging efficiency. In line with the reviewer comment, we have first changed the scales of the graphs that could have been misleading, to a more standard unit. In our experiments, the density of BrdU-positive cells is in the order of 40 cells/mm² in controls for a dose of 3*50mg BrdU/kg.

These values of cell counts are consistent with the literature. For instance, Valley et al (Front Neurosci 2009) reported around 60 cells/mm². Grelat et al (2018, PNAS) reported 10 cells/mm² of BrdU/c-fos positive cells, which represent between 20 and 50 % of all BrdU cells, yielding an extrapolation between 20 and 50 BrdU-positive cells/mm².

We agree with the reviewer that this range is lower than that of other studies using higher dosages of BrdU (higher concentration and/or more injections) (Mouret et al J Neurosci 2009, 4*75mg/kg; Reshef et al Elife 2017 4*100mg/kg for instance). However, a risk for increased cell death induced by high concentrations of BrdU was reported (Caldwell et al EJN 2005, Taupin Brain Res Rev 2007), recently confirmed by Platel et al (Elife 2019). Thus, given our interest in learning-induced modulation of adult-born cell survival, we chose to keep on using a moderate dose of BrdU in the present as in our previous works (Moreno et al. PNAS 2009, Sultan et al J Neurosci 2011.; Mandairon et al J Neurosci. 2012, Mandairon et al eLife 2018, Forest et al Cereb Cortex 2019).

- In figure 3, the authors argue that a decrease in CldU numbers in group 2 matches the decrease in performance. However, there is no significant difference between group 2 and group 1, which performs the discrimination, weakening their claim.

In addition, these are not cells activated by exposure to the odor, but cells positive for the proliferation marker. Similarly, the authors argue that “adult-born neurons are sensitive to interference only during their first 3 weeks and that if further learning occurs when adult-born neurons saved by the first are more than 21 days old, they are resistant to death and memory persists.”. However, they do not tag the neurons to demonstrate it is indeed the neurons involved in the first learning that are dying. To more strongly demonstrate this, an activity dependent tagging approach is necessary, particularly given the previous point about cell numbers.

We first would like to draw the attention of the reviewer the fact that group2 does not discriminate +lim from -lim (no difference between Hab4 and test in Figure 3Bi) while group 1 does (difference between Hab4 and test, Figure 3Bi). In line with these behavioral differences, CldU is lower in group2 compared to group1 (Figure 3Cii).

As suggested by the reviewer, in addition to proliferation marker, we now provide the cell activity marker BrdU/Zif268 double labelling over all time points in new Figure 1D and Figure 2D to cover and flank the time window during which neurons are sensitive to interference. We acknowledge we do not directly identify dying adult-born neurons as those involved in the first learning. However, we provide evidence that once BrdU positive cell density declines (Figure 1Ci, T4 and T5), a smaller proportion of adult-born respond to the learned odorant (Supplementary Figure 3, T4 and T5 and Figure 1D, T4, T5), suggesting that adult-born neurons that died responded to the learned odorant. Further evidence for their involvement in learning comes from the optogenetic experiment. In this experiment, death of adult-born neurons was prevented by keeping the odorants in the environment and their inactivation impaired learned discrimination.

We have rewritten part of the results and discussion sections to take into account the reviewer comments. In particular, we tuned down the sentence wisely pointed out by the reviewer.

- This perceptual olfactory discrimination protocol has been published several times and very successfully by this group. However, replication in a mouse strain that considerably explores odors (129/C57?), in which exploration levels - particularly between Hab4 and test - would be greater than 1 second, would make these findings significantly more convincing from a behavioral perspective.

As requested by the reviewer, we replicated the experiment with 129 mice from Charles River (see New supplementary Figure 5). We replicated the T2' experimental group and found, as for the C57Bl6 mice, that when 2 learning are separated by a short time, animals are able to remember only the second odor pair. This new experiment reinforces previous results.

Results and Methods sections have been modified accordingly.

Minor points

- Is there any overlap between CldU and IdU positive cells?

No overlap between CldU and IdU positive cells was ever found. A sentence has been added in the result section.

- Figure 4D shows a very small reduction in cell activity with their optogenetic manipulation (about 10% of cells). This needs to be justified by the authors, especially if their cell density is as low as shown in Figure 4C.

We calculated the exact percentage of reduction of Zif268 expression by EYFP cells induced by light stimulation shown in Figure 4D and found a reduction of 30.35% (control 41.94 versus Halo 29.21 in Figure 4D).

In Figure 4C, we report a density of about 100 EYFP (halo transduced) cells/mm². 40% of these are Zif268 positive in control (Fig4D), corresponding to 40 EYFP-Halo-Zif268/mm². These values closely compare to the 30 EYFP-c-fos cells/mm² reported by Alonso et al (Nature Neuro 2012) using ChR2 in

a lentivirus similar to ours. Optogenetic stimulation changed activity of about 40% in Alonso et al (against 30 % in our study) and lead to clear behavioral effect. Interestingly, modulating about 320 adult-born neurons/dentate gyrus was reported to alter hippocampal memory performance (Zhuo et al Elife 2016).

Reviewers' Comments:

Reviewer #1:

Remarks to the Author:

The authors have provided additional experiments with associated text revision to address my concerns. I am satisfied with the responses.

Reviewer #2:

Remarks to the Author:

The authors have sufficiently addressed my concerns with the clarifications and novel experiments. The addition of the Zif268 datasets and the second mouse strain strengthened their findings.

However, in the absence of a tagging approach to differentiate and properly track neurons encoding the first versus second odor exposure episodes, their conclusion that there is a direct correlation between cell death and memory encoding is not fully convincing. In fact, the new data shows that whereas the T2 interference experiment leads to a decrease in BrdU density and BrdU/Zif density, the percentage of Brdu/Zif cells is very similar (Sup figure 7, authors cite statistical difference but groups are extremely similar). However, at T4 and T5, when lim discrimination is also absent, percentage is far lower, while density is not that much lower than T2 (particularly for T4). This suggests there is more to the straightforward 1 neuron: 1 memory hypothesis the authors are putting forward.

Similarly, the reduction in CldU density is very small (Fig3), such that there are many neurons left, and there is no direct evidence that all the neurons that die are encoding of the olfactory memory (e.g. a greater effect in the percentage of CldU/zif). It seems more likely that a less straightforward process is taking place, and the authors should consider further acknowledging this complexity in the text. The authors have already made a first step in this direction, including by changing the title, but further care in acknowledging the limitations of their strategies and especially in the assertion of their conclusions would improve the manuscript and may help protect them from targeted criticism.

Reviewer #1 (Remarks to the Author):

The authors have provided additional experiments with associated text revision to address my concerns. I am satisfied with the responses.

We thank the reviewer for his/her positive response.

Reviewer #2 (Remarks to the Author):

The authors have sufficiently addressed my concerns with the clarifications and novel experiments. The addition of the Zif268 datasets and the second mouse strain strengthened their findings.

However, in the absence of a tagging approach to differentiate and properly track neurons encoding the first versus second odor exposure episodes, their conclusion that there is a direct correlation between cell death and memory encoding is not fully convincing. In fact, the new data shows that whereas the T2 interference experiment leads to a decrease in BrdU density and BrdU/Zif density, the percentage of Brdu/Zif cells is very similar (Sup figure 7, authors cite statistical difference but groups are extremely similar). However, at T4 and T5, when lim discrimination is also absent, percentage is far lower, while density is not that much lower than T2 (particularly for T4). This suggests there is more to the straightforward 1 neuron: 1 memory hypothesis the authors are putting forward.

Similarly, the reduction in CldU density is very small (Fig3), such that there are many neurons left, and there is no direct evidence that all the neurons that die are encoding of the olfactory memory (e.g. a greater effect in the percentage of CldU/zif). It seems more likely that a less straightforward process is taking place, and the authors should consider further acknowledging this complexity in the text. The authors have already made a first step in this direction, including by changing the title, but further care in acknowledging the limitations of their strategies and especially in the assertion of their conclusions would improve the manuscript and may help protect them from targeted criticism.

We have toned down our claims and conclusions in the abstract, introduction and result sections and we have added a paragraph in the discussion according to his/her comments. The last changes to the manuscript are in blue.